# Effect of disinfection agents and quantification of potentially viable Leptospira in fresh water samples using a highly sensitive integrity-qPCR assay

**Elise Richard[1,2], Pascale Bourhy[2], Mathieu Picardeau[2]\*, Laurent Moulin[1]\*, Sébastien Wurtzer[1]**

**1** Eau de Paris, DRDQE, Ivry-Sur-Seine, France, **2** Institut Pasteur, Unité Biologie des Spirochètes, Paris, France

\* mathieu.picardeau@pasteur.fr (MP); laurent.moulin@eaudeparis.fr (LM)

**Data Availability Statement:** All relevant data are within the paper and its Supporting Information files.

**Funding:** This work is part of the PhD thesis of E. R., who received financial support from the "DIM

## Abstract

Leptospirosis is an emerging worldwide zoonotic disease, but the general biology of the causative agents is still poorly understood. Humans are an occasional host. The main risk factors are water-associated exposure during professional or recreational activities or during outbreaks in endemic areas. Detecting the presence of pathogenic bacteria in aquatic environments and their capacity to resist various inactivation processes are research fields that need to be further developed. In addition, the methods used for detecting and enumerating *Leptospira* still need to be improved. We aimed to describe a new quantitative polymerase chain reaction coupled to propidium monoazide treatment (PMAqPCR) that targets not only total *Leptospira* but also discriminates pathogenic from non-pathogenic *Leptospira* while also addressing PCR inhibitors, a frequently encountered problem when studying environmental water. In a second step, the killing efficiency of *Leptospira* to different treatments was tested and PMAqPCR compared to culture-based enumeration. This provided information about the effects of temperature, as well as ultraviolet and chlorine disinfection, that are both related to water treatment processes, in particular for the production of drinking water, on the persistence of both saprophytic and pathogenic *Leptospira*. Finally, PMAqPCR was used for the detection of *Leptospira* in freshwater samples for a proof-of-concept. In conclusion, our method could be used for routine freshwater monitoring and allows better evaluation of the presence of *Leptospira*, allowing evaluation of the bacterial dynamics in a designated area or assessment of the efficacy of water disinfection processes.

## Introduction

Pathogenic *Leptospira* are responsible for a global zoonosis, leptospirosis, in which humans are found to be occasional hosts in a cycle involving wild and domestic animals. One million severe cases are reported every year worldwide [1]. Leptospirosis can take various forms, from a flu-like syndrome (fever, myalgia, or headache) to a multisystem disorder, with icteric and hemorrhagic syndrome accounting for 20% of cases, causing at least 60,000 deaths a year.

1Health 2017 from Ile-de-France region". The funders had no role in study design, data collection and analysis, decision to publish, or preparation of the manuscript.

**Competing interests:** The authors have declared that no competing interests exist.

This disease occurs worldwide but the incidence is the highest in tropical regions [1–3]. However, developed countries, including those in Europe [4, 5], have also observed an increase in the number of reported cases.

Animal reservoirs, mainly rodents, excrete *Leptospira* through urine and contaminate the environment and, potentially, water resources. The dissemination of these bacteria into the environment can allow other animals or humans to be newly infected. Humans can be infected through direct or indirect contact with urine or water contaminated by *Leptospira*. In this context, leptospirosis can be both an occupational disease (affecting veterinarians, farmers, sewer workers, etc.) and a recreational disease associated with water-related activities (bathing, kayaking, canyoning, etc.) [6, 7]. Leptospirosis is considered to be an emerging zoonotic disease partially due to global warming [8] and more frequent and severe flooding events [9]. Water exposure appears to be the major risk factor [10]. Floods increase the risk by exposing humans and animals to *Leptospira* that are flushed out of their environment. In France, a study conducted from 1995 to 2005 showed that 42% of patients became infected after practicing water sports, 19% after contact with backwater (ponds, swamps, wells, water holes), and 19% during professional activities [11]. Recreational water activities are becoming increasingly popular and, for example, a recent study described a cluster of 14 kayakers that exhibited leptospirosis symptoms after contact with water in Britanny, France [12]. Leptospirosis cases have also been reported after the consumption of drinking water. Contamination may be caused by failure during water treatment processes or the lack of any water treatment [13, 14].

To date, sixty-four species and more than three hundred serovars of *Leptospira* have been described and classified into four phylogenetic subclades: pathogenic P1 and P2 and saprophytic S1 and S2 [15]. Saprophytes are non-infectious species that can multiply in the environment, whereas pathogens are mostly isolated from both humans and animals and, occasionally the environment, in which they can survive for a few weeks [16]. *Leptospira* are slow-growing bacteria (generation time of 5 and 20 hours for saprophytes and pathogens, respectively), requiring a specific and rich medium [17, 18]. They are fastidious to isolate because of possible contamination with fast-growing interfering flora [19–21]. The recent development of a cocktail of antibacterial and antifungal antibiotics to which *Leptospira* are resistant (sulfamethoxazole, trimethoprim, Fosfomycin, 5-fluorouracil, amphotericin B,), thus limiting the development of interfering microorganisms [22], should facilitate their culture.

Little information is available on *Leptospira* contamination of surface water and its seasonal evolution. Several authors have highlighted the importance of assessing the bacterial concentration in water resources [19]. Such evaluation is essential for monitoring population exposure to *Leptospira* to ensure public health. Currently, scarce data are available about the persistence of *Leptospira* in the environment and their resistance to physico-chemical parameters or disinfection. According to Huang et al. [23], *Leptospira* genomes were still detected in tap water after treatment, whereas most other pathogens disappeared.

The effects of temperature and pH on *Leptospira* were studied by Chang et al. [24]. In this study, the authors showed that optimal survival conditions for *Leptospira* are neutral to slightly alkaline pH and a temperature of approximately 25 to 27˚C. Other studies [25, 26] showed that pathogenic strains of *Leptospira* could survive for > 20 months, despite deleterious storage conditions (cold, nutrient-poor acidic water, etc.). To date, no study has described the antimicrobial effect of ultraviolet (UV) radiation exposure, a process commonly used in tap water production, on the viability of *Leptospira*.

Molecular diagnostic methods appear to be more sensitive, specific, and rapid than culture [27, 28]. qPCR appears to be applicable for *Leptospira* detection, although there is no consensus concerning molecular methods. In most recent studies, real-time PCR assays have been based on Taqman technology [29, 30] instead of the SYBR Green approach [27, 31] due to its

higher specificity [29]. Molecular methods for *Leptospira* detection have been mainly based on the detection of housekeeping genes, such as *rrs* 16S [32], *gyrB* [33], or *secY* [27]. Pathogen-specific *Leptospira* can be detected using genes such as *LipL32* [29], *ligA*, or *ligB* [34]. However, most analyses solely allow quantification of the *Leptospira* genome, irrespective of the viability of the bacteria. These methods tend to overestimate the true risk, which is solely linked to viable bacteria. The use of qPCR coupled with a DNA intercalating agent pretreatment was already described to evaluate the integrity of *Leptospira* [35].This approach avoids the amplification of permeable bacteria (i.e. damaged or dead bacteria) and, thus, partially limits their overestimation while maintaining the advantages of molecular methods [36].

Our objective was to create a rapid and sensitive method for the quantification of potentially viable *Leptospira* in water samples, discriminating pathogens from saprophytes. This PMAqPCR detection method was also used to evaluate the resistance of *Leptospira* to different treatments such as heat inactivation, chlorine treatment, and UV exposure.

## Materials and methods

### Sample collection and culture media

Twenty-five *Leptospira* DNA samples, including that of eight saprophytes (four for S1 and four for S2) and seventeen pathogens (twelve for P1 and five for P2) (S1 Table) were used to test the performance of the PCR assays. The pathogen *L. interrogans* serovar Manilae strain L495 and the saprophyte *L. biflexa* serovar Patoc strain Patoc 1 were used for inactivation tests. Strains were incubated at 30°C in liquid Ellinghausen, McCullough, Johnson, and Harris (EMJH) medium [37, 38].

For colony numeration, EMJH medium supplemented with 1.2% agar was used. Bacterial numbers were determined using a Petroff-Hausser chamber and dark-field microscopy. *Leptospira* DNA and strains were provided by the National Reference Center (NRC) for Leptospirosis (Institut Pasteur, Paris, France). Other bacterial strains used to determine the specificity of the pan-*Leptospira* PCR assay were provided by the Eau De Paris Laboratory.

### Extraction

For inactivation tests, DNA was extracted using the Q400 protocol with a QIAsymphony® DSP Virus/Pathogen Midi Kit (Qiagen) and the MagNA Pure Compact System (Roche®) for environmental monitoring. Nucleic acids were extracted from 200 μL of sample by elution into 50 μL, according to the manufacturer's protocols.

### PCR assays

The multiplex qPCR was tested using *Leptospira* strains isolated from patients and environmental strains. The specificity of the qPCR methodology was established by the analysis of *Leptospira* strains and other bacterial species (S1 Table).

The Pan-*Leptospira* PCR was based on the *rrs* (16S) gene sequence alignment of 113 *Leptospira* species, including saprophytes and pathogens. The primers 16S-F267 (`5'-GGCCACAATGGAACTGAG-3'`) and 16S-R336 (`5'-CCCATTGAGCAAGATTCTTAAC-3'`), associated with the probe 16S-P286 (`5'- FAM-CACGGTCCATACTCCT-NFQ-MGB-3'`), achieve the amplification of a 70-bp fragment. The pathogenic-*Leptospira* PCR was based on the *LipL32* gene sequence alignment of 30 pathogenic *Leptospira* species. The primers LipL32-F164 (`5'-CTGTGATCAACTATTACGG-3'`) and LipL32-R298 (`5'-GGGAAATCATACGAACTC-3'`), associated with the probe LipL32-P188 (`5'-HEX-TAAAGCCAGGACAAGCGCCG-BHQ1-3'`), achieve the amplification of a 135-bp fragment.

Sequences from the Genbank database of the National Center for Biotechnology Information (NCBI) were used to design the primers (S2 and S3 Tables). The primers and probes for both PCR assays were designed using AlleleID® software version 7 (http://premierbiosoft.com/bacterial-identification/index.html).

Two plasmids (16S and *LipL32*) were generated as positive PCR controls using pCR2.1 (Topo TA-cloning, Life Technologies, Carlsbad, CA). An internal positive competitive amplification control (IPC) was used [39] to evaluate the presence of PCR inhibitors. The IPC is composed of a partial sequence of the human *β-actin* gene and was cloned into the pCR™2.1-TOPO® vector (Life Technologies, Carlsbad, CA) and flanked by the *LipL32* primers using an approach similar to that described by Wurtzer et al. [40]. The primers were LipL32_BACT-F1146 (5′-CTGTGATCAACTATTACGGttGCAGGAGTATGACGAGT-3′) and LipL32_BACT-R1215 (5′-GGGAAATCATACGAACTCttCAAGAAAGGGTGTAACGCAACTAA-3′). The probe was BACT_P1172 (5′-CCCCTCCATCGTCCACCGCAAATG-3′).

Each PCR reaction was performed using the TaqMan ™ Fast Virus 1-Step Master Mix (#4444434). Unlike other polymerases, this Taq polymerase was less affected by inhibition due to the environmental sample matrix. The RT step was removed to achieve rapid diagnosis. Nevertheless, residual activity of the RT step remained, thus conferring better sensitivity than other kits.

For the pan-*Leptospira* PCR (16S), the F267 primer was used at 500 nM and the R336 primer and P286 probe at 100 nM. Simultaneously, oligonucleotides targeting pathogenic-*Leptospira* PCR (*LipL32*) were used at 600 nM for primer F164 and primer R298, and 200 nM for probe P188. IPC was added to the reaction mixture at $10^4$ copies and detected using the IPC probe at 100 nM.

The PCR reaction was performed in a 20-µL reaction volume using a ViiA 7 real-time PCR system (Life Technologies, Carlsbad, CA) in 96-well plates. The thermal profile consisted of an initial denaturation step at 94˚C for 20 s, followed by 45 cycles at 94˚C for 5 s and 60˚C for 30 s. FAM Yakima Yellow and Tamra fluorescence were detected at the end of the elongation step.

The three plasmids, positive controls, and IPC were quantified using an ultra-sensitive fluorescent nucleic acid stain for quantitating double-stranded DNA (Quant-iT™ PicoGreen® dsDNA reagent), according to the manufacturer's protocol.

## Propidium monoazide treatment

Before DNA extraction, samples were incubated with propidium monoazide (PMAxx) to ensure bacterial integrity [41–43]. The PMAxx solution was diluted in molecular grade water to obtain a final concentration of 10 mM and aliquots were stored at -20˚C. Based on pilot studies (data not shown), the PMAxx dye was used at a final concentration of 0.1 mM to pre-treat the samples. After mixing, samples were incubated on ice, in the dark, for 30 min. Photo-activation was performed for 15 min using the PhaST Blue system (IUL, Barcelona, Spain). In this study, PMAxx-qPCR was also called integrity qPCR and is sometimes wrongly called viability PCR.

## Persistence tests

Persistence tests were simultaneously performed in duplicate on two laboratory strains: *Leptospira biflexa* serovar Patoc (saprophyte) and *Leptospira interrogans* serovar Manilae (pathogen).

For each test, bacterial inactivation was modeled using GraphPad Prism version 8 software (GraphPad, La Jolla, CA). After testing several models, the *Leptospira* reduction data were adjusted using a sigmoidal dose−response model based on the equation:

$$Y = Ct + \frac{(C_0 - Ct)}{1 + \left(\frac{IC_{50}}{X}\right)^{\alpha}}$$

In this equation, X was the studied parameter (temperature, CT value, time, UV dose), the variable "Ct" the *Leptospira* concentration at a given time during the assay, "$C_0$" the *Leptospira* concentration at T0, and IC50 the value of X at which the response was halfway between $C_0$ and C0. Finally, α described the slope of the curve.

As mentioned in every test referenced below, two detection methods were used in this study: integrity qPCR (described above) and culture on specific EMJH medium.

**Heat exposure.** Bacterial cultures were adjusted to a final concentration of approximately $10^4$ *Leptospira*/mL in 1X PBS (pH 7.4). Each sample was aliquoted in duplicate and incubated for 1 h at various temperatures in a thermal cycler (Mastercycler® nexus, Eppendorf). Samples tested at 4˚C were stored on ice.

Half of each sample was used for plating after resuspension in EMJH medium and half for PMA-qPCR analysis after resuspension in 1X PBS and the addition of PMAxx (100 μM final concentration). Nucleic acids were extracted using a QIAsymphony instrument (QIAGEN).

**Ultraviolet (UV) exposure.** Exponential phase cultures of *L. interrogans* and *L. biflexa* were centrifuged at 8,000 x g for 10 min and the pellet resuspended in 1X PBS to a final concentration of $10^9$ *Leptospira*/mL. The bacterial suspension was split into microtubes (450 μL/tube). For each strain, eight UV conditions were tested in duplicate, from 0 mJ/cm2 to 40 mJ/cm$^2$ (the last is the UV dose applied in drinking-water treatment plants) using a 254 nm UV lamp (Phillips, Amsterdam, Netherlands) at room temperature. A digital UVC radiometer (IL Metronic Sensortechnik GmbH, Germany) was used to monitor UV irradiation.

Samples were subjected to various exposure times according to the tested dose and half used for plating on EMJH agar and half for PMA-qPCR analysis.

**Chlorine treatment.** Before starting the experiments, laboratory glassware was prepared by soaking it in a sodium hypochlorite solution containing 40 mg/L free active chlorine. Glassware was then intensively cleaned with chlorine demand-free (CDF) water. CDF water was used to prepare all experimental solutions and was prepared using a Milli-Q® Purification system with a Biopak® Polisher (Merck, Darmstadt, Germany). A stock solution of sodium hypochlorite (81 g/L) was used to prepare an intermediate solution of 0.5 mg/L with CDF water (pH 6).

As described previously, *L. interrogans* and *L. biflexa* were incubated at 30˚C in liquid EMJH medium until reaching a concentration of approximately $10^8$ *Leptospira*/mL. *Leptospira* cultures were centrifuged at 5,000 x g for 15 min, the supernatant discarded, the pellet washed in 0.9% NaCl, and finally resuspended in 0.9% NaCl. *Leptospira* suspensions were prepared by adding bacteria (final concentration of $10^4$ *Leptospira*/mL) to a chlorine solution (0.5 mg/L). The free chlorine concentration was measured before and after addition of the *Leptospira* suspension using a Pocket colorimeter II (Hach Lange, Dusseldorf, Germany) after activation of the DPD reagent (N,N-diethyl-p-phenylene-diamine).

Unreacted free chlorine was quenched by the addition of sodium thiosulfate (100 mg/L) to stop the activation reaction. After chlorine treatment, each sample was analyzed by culture, qPCR, and integrity qPCR.

## Environmental sample collection

Thirty-four surface-water samples were collected in Paris, including an area which includes a controlled bathing section during the summer period (48.885441128096, 2.37411186180 21346). The surface water had not undergone any sanitation treatment.

Samples were analyzed within 24 h of collection. Surface-water samples (500 mL) were concentrated by successive centrifugation steps down to 400 μL. Half of the sample was treated with PMAxx (as described in the section on propidium monoazide treatments) and half remained untreated. All samples (with and without PMAxx) were then extracted and analyzed by qPCR.

### Statistical analysis with GraphPad software

In addition to its use in the persistence test to model bacterial inactivation, GraphPad Prism software version 8 (GraphPad, La Jolla, CA) was also used for statistical analysis. Normality of the distribution was assessed using the Shapiro-Wilk test. Paired groups were tested using the nonparametric Wilcoxon matched pairs signed rank test. P-values < 0.05 were considered significant.

## Results

### *Leptospira* multiplex qPCR

First designed to operate separately, the three qPCR assays were adapted to be performed in a multiplex assay to obtain a single reaction (effectiveness demonstrated below in the section on the analytical sensitivity of the method). This was made possible through the use of different fluorophores. Total *Leptospira*, pathogenic *Leptospira*, and IPC were targeted by different dyes: FAM, Yakima Yellow, and TAMRA, respectively.

Several enzymatic master mixes were tested to optimize the qPCR reaction (Fig 1). The Fast Virus 1-Step Master mix resulted in better detection limits than amplification using two other enzyme mixes, with a difference of approximately 10 CT between qPCRs. In addition, the efficiency and coefficient of determination (R2) showed the Fast Virus 1-step master mix to outperform the others in this study (Table 1).

As indicated previously, the two qPCR assays for detecting *Leptospira* and the IPC were first designed to operate independently. We compared the performance of the qPCR assays separately (simplex mode) or together (multiplex mode). The simplex and multiplex modes provided the same results for the 16S target (efficiency of approximately 100% and R2 of approximately 0.98) (Fig 2) and these two parameters were slightly higher in the multiplex mode (89% of efficiency of 89% and R2 of 0.987) than the simplex mode (efficiency of and 84% R2 of 0.984) for the *LipL32* target (Table 2).

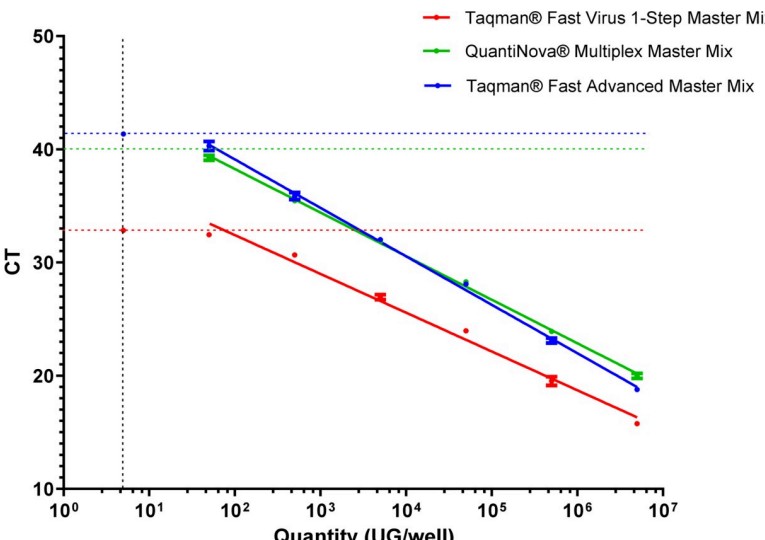

**Fig 1. qPCR tests on range of concentrations of the Leptospira 16S plasmid with various enzymatic mixes (Taqman® Fast Virus 1-Step Master Mix, QuantiNova® Multiplex Master Mix, Taqman® Fast Advanced Master Mix).** The colored dotted lines correspond to the number of cycles required to obtain the smallest detectable quantity of genome, shown by the black dotted line.

**Table 1. Comparison of $R^2$ and efficiency between various enzymatic mixes.**

|  | TaqMan® Fast Virus 1-Step Master Mix | QuantiNova® | Taqman® Fast Advanced Master Mix |
|---|---|---|---|
| $R^2$ | 0.984 | 0.996 | 0.994 |
| Efficacy (%) | 95.8 | 81.9 | 71.3 |

The efficiency and $R^2$ were compared between various enzymatic mixes from two suppliers: TaqMan® (Applied Biosystems) and QuantiNova® (Qiagen).

Ten-fold serial dilutions (ranging from $10^6$ copies/μL to $10^3$ copies/μL), followed by two-fold serial dilutions (from $10^3$ copies/μL to 1 copy/μL), of *L. interrogans* serovar Manilae were used to determine the true analytical sensitivity of the detection method. The limit of detection (LoD) was determined as the quantity of plasmid that could be detected in 95% of the replicates. The limit of quantification (LoQ) was the lowest concentration of plasmid that could be properly quantified in a standard range. The LoD, LoQ, and amplification range were separately determined for each target (16S, *LipL32*, and IPC) to determine their own parameters. The LoD and LoQ of 16S were both at a CT value of 33.09, corresponding to 1 bacterium/well. For *LipL32*, the LoQ was at a CT value of 37.97, corresponding to 125 bacteria/well (Fig 3), and the LoD at 38.66, corresponding to 86 bacteria/well.

We tested the specificity of the pan-*Leptospira* PCR on bacterial strains other than *Leptospira*. The assay was found to be specific for *Leptospira* spp., as none of the six other pathogenic organisms were amplified. Moreover, we tested the specificity of pathogenic-*Leptospira* PCR. None of the nonpathogenic *Leptospira* were detected by the assay. The sensitivity was also measured, and results were mentioned in S1 Table.

## Evaluation of the efficacy of disinfection treatments of *Leptospira*

Addition of the integrity assay to the detection by qPCR enabled the specific detection of unaltered bacteria and viable but non-cultivable (VBNC) bacteria in the samples [44]. An intercalating agent was added and photoactivated to avoid the amplification of "free" or not protected DNA by the qPCR. This protocol was tested on *Leptospira* subjected to disinfection treatments (temperature, UV radiation, chlorine). The bacterial concentration was evaluated by three different methods to validate the use of PMAqPCR: microscopic enumeration, colony-forming units by plating, and PMAqPCR (Table 3).

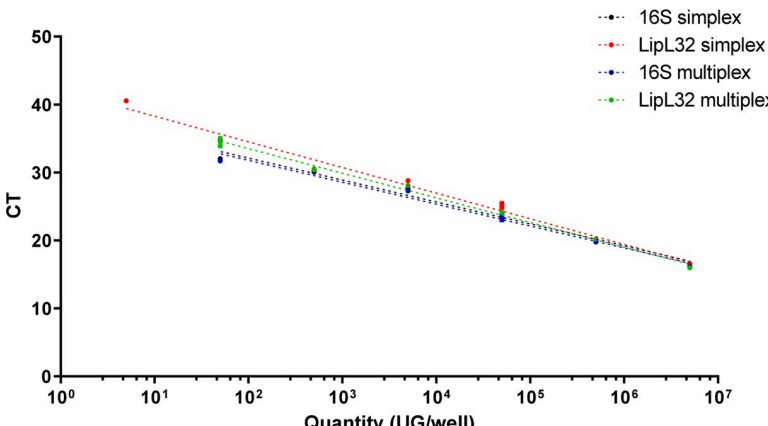

**Fig 2. Comparison between simplex and multiplex qPCR for the two targets: 16S gene and LipL32 gene; using range of plasmids.**

**Table 2. Comparison of R² and efficiency between the simplex and multiplex qPCR assays developed in this study.**

| | *rrs* (16S) simplex | *LipL32* simplex | *rrs* (16S) multiplex | *LipL32* multiplex |
|---|---|---|---|---|
| **R²** | 0.9827 | 0.9846 | 0.9820 | 0.9873 |
| **Efficacity (%)** | 104 | 84 | 104 | 89 |

The efficiency and R² were compared between the simplex and multiplex modes for two targets: rrs (16S) gene for all Leptospira and LipL32 gene for pathogenic strains.

The titrations obtained by microscopy and plating were similar, giving approximately $10^7$ *Leptospira*/mL. Quantification of the two genes indicated concentrations 25 to 60 times higher.

Various experiments were independently carried out with fresh *Leptospira* suspensions. Thus, the initial concentration of bacteria could have differed. Nonetheless, this parameter did not interfere with the interpretation of results because the analysis was based on the log of inactivation.

Heat inactivation was assessed by culture analysis and PMA-qPCR analysis. The pathogenic strain appeared to be more resistant than the saprophytic strain. Indeed, the curve of the slope from the culture analysis was lower for *L. interrogans* (-6.945) than *L. biflexa* (-8.211). However, the temperature which induced a reduction of the concentration of the bacteria by half was broadly similar for both strains (32.12˚C for *L. interrogans* and 32.91˚C for *L. biflexa*) (Fig 4).

*Leptospira* appeared to not be cultivable on EMJH solid plates beyond 37˚C and not detectable by PMA-qPCR beyond 55˚C.

The time of exposure to free chlorine was calculated from the kinetics of free chlorine consumption and adjusted to reach a CT value equal to 10 mg.min/L (Fig 5). This CT value corresponded to the concentration of this powerful oxidant (mg/L) multiplied by the time (min) of exposure. In this case, we selected the time at which the area under the free chlorine CT curves

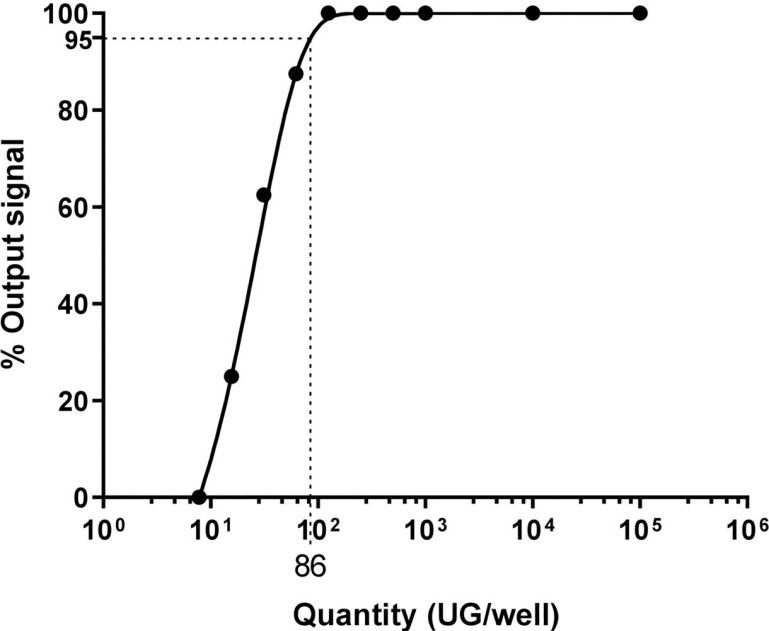

**Fig 3. Determination by PMAqPCR of LoD for the LipL32 gene using serial dilution (eight replicates for each dilution).**

**Table 3. Comparison of *Leptospira* detection between microscopy, plating and molecular analysis.**

| | Microscopic enumeration | Plating enumeration (EMJH) | | PMAqPCR (PBS) | PMAqPCR (EMJH) |
|---|---|---|---|---|---|
| | n = 9 | n = 9 | | n = 6 | n = 6 |
| *L. biflexa* Patoc | 1.00E+07 | 1.66E+07 | (16S) | 4.25E+08 | 4.21E+08 |
| *L. interrogans* Manilae | 1.00E+07 | 1.63E+07 | (16S) | 1.11E+09 | 7.10E+08 |
| | | | (*LipL32*) | 7.45E+08 | 4.65E+08 |

Microscopic enumeration was assessed by dark field microscopy using a Petroff-Hausser chamber. Plating enumeration was performed using EMJH semi-solid medium after seven days of incubation for *L. biflexa* and more than two weeks of incubation for *L. interrogans*. Detection by molecular analysis was performed in PBS and EMJH liquid medium.

was 10 mg.min/L. The time of exposure ranged from 0 to 27 min. Chlorine-dependent inactivation was assessed by culture and molecular methods. We observed a slight decrease in the concentration of *L. interrogans* in the presence of the hypochlorous acid during the assay by PMAqPCR. The effect was stronger for *L. biflexa*, with 1.5 log removal at CT = 4mg.min/L and up to 2 logs of inactivation at the end of the experiment (CT = 10mg.min/L). Both strains showed strong sensitivity to chlorine treatment by culture assay (Fig 6). No *L. interrogans* grew on the EMJH culture media after exposure to the lowest chlorine dose (CT = 0.001 mg.min/L), whereas total inactivation of *L. biflexa* occurred at CT = 0.1 mg.min/L.

UV$_{254}$ light inactivates microorganisms by targeting their nucleic acids, resulting in the inhibition of DNA replication and thus their growth in culture. Culture analysis showed the same kinetics for the action of UV radiation for both strains (Fig 4). Three logs of removal were achieved at 10 mJ/cm$^2$, whereas 40 J/cm$^2$ is currently used in drinking water treatment plants. UV light kills cells by damaging their DNA and does not usually result in cell lysis; the

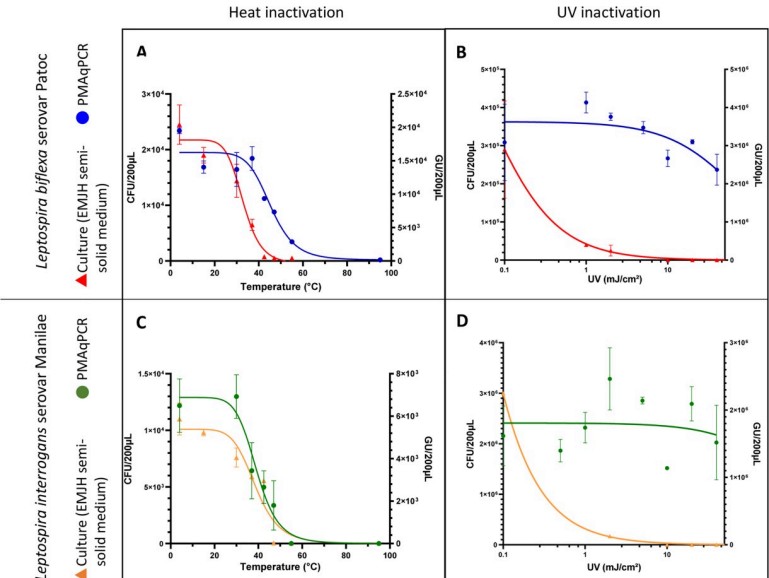

**Fig 4. Heat and UV inactivation for *Leptospira biflexa* serovar Patoc (A&B, respectively) and for *Leptospira interrogans* serovar Manilae (C&D, respectively).** The X-axis indicates the temperature to which *Leptospira* were exposed for 1 h (A, C) or the UV dose of exposure (B, D). The left Y-axis represents the median bacterial culture results (in colony-forming units) after several days of incubation in EMJH semi-solid medium at 30˚C (circle). The right Y-axis represents the median PMAxxqPCR results (in genome units) after extraction and molecular biology assay based on the 16S gene (triangle).

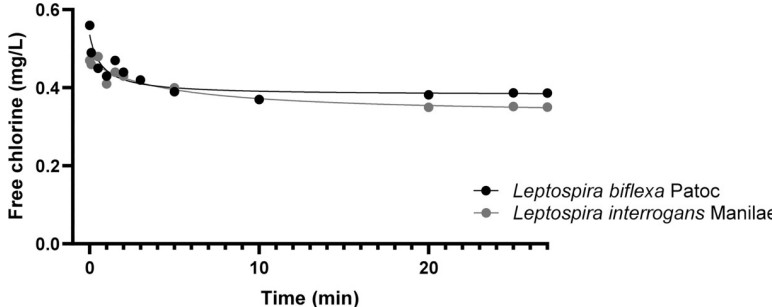

**Fig 5. Free chlorine consumption kinetics for *L. biflexa* serovar Patoc and *L. interrogans* serovar Manilae.** The time of exposure to free chlorine was adjusted to maintain a CT-value equal to 10 mg.min.$L^{-1}$.

lack of decrease for PMAqPCR signal is therefore consistent with the preservation of membrane integrity.

**Proof-of-concept of application for environmental monitoring.** Our PMAxx-qPCR method was tested on 32 surface water samples, with and without PMAxx. The results are summarized in Fig 7. A Wilcoxon signed-rank test was performed to show whether the data with PMAxx were significantly different from those without. The two sets of data were significantly different (p-value < 0.0001). The genome concentration using PMAxx-qPCR was consistently lower than that estimated using conventional qPCR in 100% of cases, signifying that a significant proportion of target DNA was "free" or inside permeable bacteria. Adding PMAxx reduced overestimation by amplifying only the genomes of non-permeable bacteria (potentially viable bacteria).

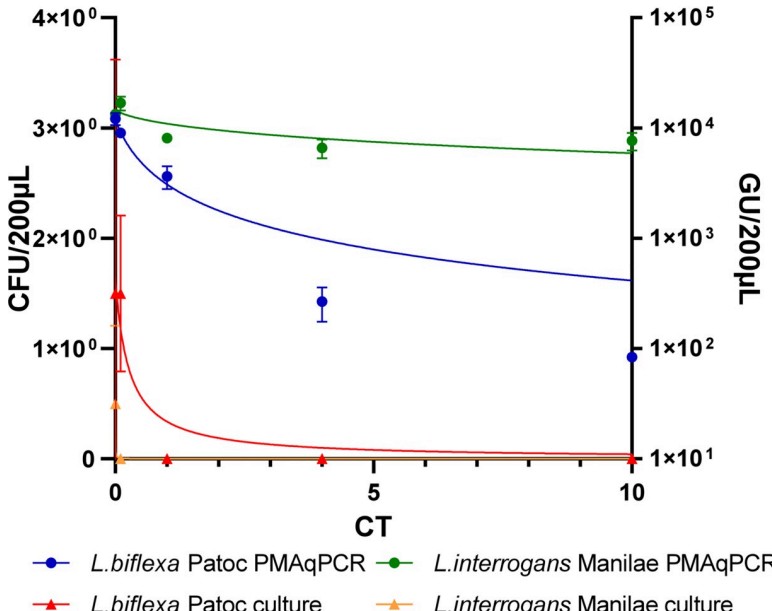

**Fig 6. Chlorine inactivation for *L. biflexa* serovar Patoc and *L. interrogans* serovar Manilae by culture and PMAqPCR.** Results were gathered for the two strains *L. biflexa* serovar Patoc (blue, red) and *L. interrogans* serovar Manilae (green, orange). The X-axis indicates the CT value: powerful oxidant concentration (mg/L) * time (min) of exposure. The left Y-axis represents the median bacterial culture results (in colony-forming units) after several days of incubation in EMJH semi-solid medium at 30˚C (triangle). The right Y-axis represents the median PMAxxqPCR results (in genome units) after extraction and molecular biology assay based on the 16S gene (circle).

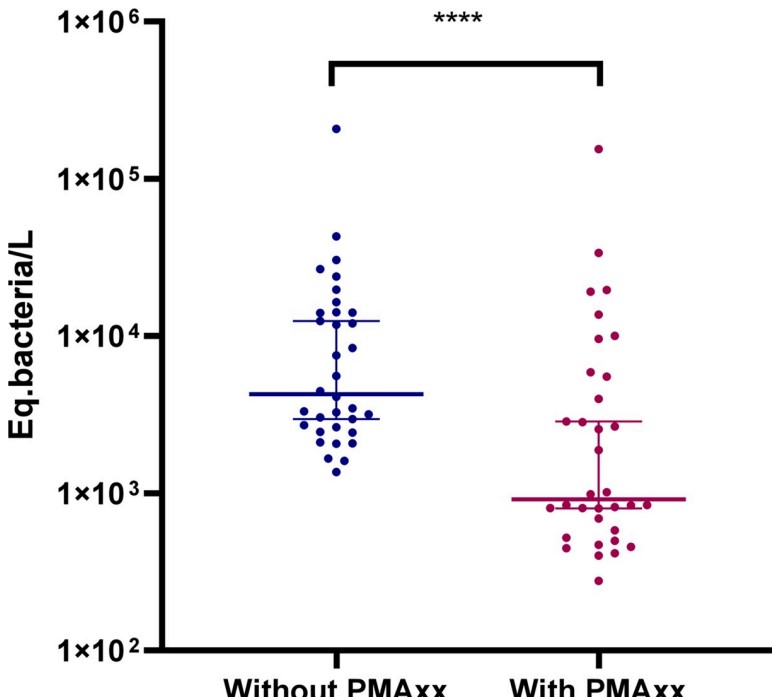

**Fig 7. Distribution and median of *Leptospira* genome (16S gene) with (red) and without PMAxx (blue) on environmental samples.** This analyze was based on 34 environmental freshwater samples collected in the Ourcq canal (Paris) in June-September 2018.

## Discussion

Leptospirosis is an emerging waterborne zoonosis of global importance for both humans and animals. However, there is also an urgent need for a robust and easy-to-use *Leptospira* detection method for environmental samples. Traditional culture methods are fastidious and Leptospira are slow-growing bacteria and cultures can be contaminated by other microorganisms [28]. Alternative detection methods, such as specific qPCR, have thus been proposed but no gold-standard has been implemented for environmental monitoring. In general, molecular methods allow a rapid measurement of the pathogen concentration coupled with high sensitivity and specificity. In the present study, we implemented a triplex qPCR based on Taqman technology for the detection and quantification of both pathogenic and saprophytic *Leptospira* spp. using the *rrs* gene and specifically pathogenic strains based on *LipL32* gene amplification. An internal competitive amplification control was added to the analysis to evaluate the inhibition of amplification resulting from the samples. In addition, we coupled this method with a bacterial integrity assay. A triplex PCR (two genes and an internal control) was designed and used to discriminate *Leptospira* from subclades P1 and P2 in animal samples [45]. This multiplex qPCR was set up, according the MIQE guidelines [46], to detect all *Leptospira* and selectively discriminate between pathogenic and saprophytic strains. An internal control was also added as a supplementary monitor to check for inhibition of amplification in environmental samples.

Genome amplification by qPCR also has certain limitations. Amplification of any targeted DNA present in the sample makes it impossible to distinguish between live and dead cells, resulting in potential overestimation of the bacterial concentration and the risk of infection. It was also shown that the use of propidium iodide to distinguish between living bacteria from dead bacteria was not relevant under certain experimental conditions and in particular to estimate the effect of chlorination [47]. Here, we used propidium monoazide (PMAxx) to avoid

the amplification of DNA from degraded cells, permeable to the dye, and solely amplify DNA from potentially viable bacteria [35] to reduce potential overestimation.

Validation of the qPCR method was performed by comparing the molecular results (using PMAxx coupled with the RT qPCR measurement) with culture-based methods. A higher number of *Leptospira* was counted by the qPCR. This difference can be explained by various factors, such as colonies not issued from one bacteria (i.e. aggregated bacteria), genome multiplication before cells separation, count of viable but non cultivable bacteria and multiple copies of the 16S gene in *Leptospira* [48].

We compared the survival of pathogenic and saprophytic strains of *Leptospira* in several environments. Heat inactivation experiments revealed different kinetics between pathogenic and saprophytic strains, the pathogenic strain being more tolerant to heat. This relative tolerance may explain their higher prevalence in tropical areas as well as the role of these strains in human infections. Other inactivation processes did not show any significant differences in inactivation kinetics between the two strains.

Our results based on bacterial culture methods suggest that *Leptospira* is rapidly inactivated by free chlorine. However, the use of PMAqPCR shows that leptospiral membranes would not be directly damaged by free chlorine. Several studies have shown that free chlorine inactivates *E. coli* without damaging its cell membranes [49–52] further indicating that the use of PMAqPCR is not appropriate to determine the capacity of chlorine to kill bacteria.

In absence of impact of disinfection treatment, it was essential to also consider that the absence of cultivability does not necessarily indicate cell lysis as bacteria could remain as a "viable but non-cultivable" state [53].

Contrary to classical PCR, the integrity PCR approach allowed the use of more adaptable and faster molecular methods to assess inactivation efficiencies. Due to interfering flora or organic matter, the culture is too complicated to implement, especially on complex samples whose matrix negatively impacts the re-cultivation of *Leptospira*.

Certain physicochemical parameters (for example, salty water) can alter the survival of *Leptospira* [24], whereas other parameters can provide a protective effect. For example, the presence of organic matter or biofilms could increase the survival of *Leptospira* survival in aquatic environments [16, 54].

The PMAxx approach showed certain limitations concerning its use to evaluate the efficiency of disinfection treatments of *Leptospira*. Although useful results were obtained after heat or low-level free chlorine exposure, the use of PMAxx was not informative for the analysis of UV treatment. It is possible that treatments or conditions that affect bacterial integrity (temperature, chlorine) allow PMAxx to access the genome, contrary to UV radiation. However, when only DNA was targeted for the inactivation of the microorganism, PMAxx had no effect in improving the determination of *Leptospira* sensitivity (Fig 7). Such an observation has already been reported for a virus assay [55].

To date, *Leptospira* are not considered when investigating microorganisms in water. Our methods allowed us to obtain information on the presence and integrity of such bacteria in Parisian surface-water samples (Fig 7). This proof of concept should be applied to answer other questions, such as the influence of seasonal variations or the impact of rodent control campaigns. The development of a sensitive qPCR method using a rapid reverse-transcription step targeting the *rrs* (16S) or *LipL32* genes improved the sensitivity of detection [56]. Within the sampling area, the median concentration was approximately $10^3$ eq. bacteria/L. Further studies are under way to determine whether it would be relevant and useful to routinely use this method to monitor *Leptospira* in surface water. These results could be considered to establish threshold alerts, leading to restrictions of access, after events that favor bacterial contamination (heavy rain, flooding, etc.). The impact of

various parameters, such as seasons, climatic conditions, and human activities, on *Leptospira* dynamics needs to be evaluated.

Analysis of the presence of viable *Leptospira* in the environment and the measurement of the effectiveness of treatments are essential. The advantages of combining PMAxx addition and RT qPCR methods to detect low levels of non-permeable pathogens in water is now accepted [57]. The use of PMAxx combined with qPCR is therefore one possible solution for *Leptospira* measurement; it is time saving and avoids overestimation. By detecting non-permeable and VBNC bacteria, this method pinpointed and prevented a *Leptospira* outbreak originating from environmental water [58]. These data could be useful for quantitative microbiological risk assessment (QMRA) approaches.

Although it is now well known that these bacteria can have a high impact on public health in endemic areas [59] or during specific seasons [60, 61], little is known about the environmental concentration in urbanized areas in Europe affected by the proliferation of rodents, which are the main animal reservoir of *Leptospira*. With ongoing social changes, water-related activities have increased in these regions, with the installation of urban beaches, bathing areas, and aquatic activities in areas with non-treated surface water. Our PMAqPCR method will be further used to better evaluate the presence of pathogenic *Leptospira* in bathing areas in Paris, France. The monitoring of microbial contamination is a requirement for establishing microbial risk assessment guidelines. Moreover, regular monitoring of *Leptospira* could help to provide a better description of infection events. Despite an increase in reported cases of leptospirosis and evidence that most cases are due to exposure to contaminated water, regulations are still based on fecal indicators and do not yet include pathogenic *Leptospira*, probably due to the absence of reliable measurement methods. Indeed, only *Escherichia coli* and intestinal enterococci are analyzed in France (French Public Health Code-D. 1332-15D1332-15). The only existing recommendations related to the risk of *Leptospira* exposure are to avoid bathing with skin lesions or in uncontrolled bathing areas. *Leptospira* monitoring could be implemented in freshwater swimming facilities to improve awareness of *Leptospira* exposure. This could consist of identifying sources of pollution prone to affect water quality and the health of bathers. By detecting *Leptospira* in the environmental water during flooding, this assay can also contribute to early warning of potential outbreaks of leptospirosis.

## Supporting information

**S1 Table. Microorganism strains used for specificity tests and results from the TaqMan real-time multiplex (LipL32 and 16S) PCR assays.**
(DOCX)

**S2 Table. Leptospira sequences and their NCBI accession numbers required to design the primers and probe for the 16S qPCR.**
(DOCX)

**S3 Table. Leptospira sequences and their NCBI accession numbers required to design primers and probe for the LipL32 qPCR.**
(DOCX)

## Acknowledgments

We are very grateful to the sampling department of Eau de Paris for providing samples to the laboratory. We thank the staff of the National Reference Center for Leptospirosis for support and the processing of some of the samples.

## Author Contributions

**Conceptualization:** Mathieu Picardeau, Laurent Moulin.

**Formal analysis:** Elise Richard.

**Funding acquisition:** Mathieu Picardeau, Laurent Moulin.

**Investigation:** Elise Richard, Sébastien Wurtzer.

**Methodology:** Elise Richard, Pascale Bourhy, Sébastien Wurtzer.

**Supervision:** Mathieu Picardeau.

**Validation:** Laurent Moulin, Sébastien Wurtzer.

**Writing – original draft:** Elise Richard.

**Writing – review & editing:** Mathieu Picardeau, Laurent Moulin, Sébastien Wurtzer.

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
