## [Decision Letter · Decision Letter 0]

21 Apr 2021

PONE-D-21-09857

Effect of disinfection agents on Leptospira in water using a high sensitivity integrity-qPCR assay

PLOS ONE

Dear Dr. Moulin,

Thank you for submitting your manuscript to PLOS ONE. After careful consideration, we feel that it has merit but does not fully meet PLOS ONE’s publication criteria as it currently stands. Therefore, we invite you to submit a revised version of the manuscript that addresses the points raised by each of the reviewers during the review process.

We look forward to receiving your revised manuscript.

Kind regards,

Odir Antonio Dellagostin

Academic Editor

PLOS ONE

Journal Requirements:

Reviewers' comments:

Reviewer's Responses to Questions

**Comments to the Author**

1. Is the manuscript technically sound, and do the data support the conclusions?

Reviewer #1: Yes

Reviewer #2: Yes

2. Has the statistical analysis been performed appropriately and rigorously? 

Reviewer #1: Yes

Reviewer #2: Yes

3. Have the authors made all data underlying the findings in their manuscript fully available?

Reviewer #1: Yes

Reviewer #2: Yes

4. Is the manuscript presented in an intelligible fashion and written in standard English?

Reviewer #1: Yes

Reviewer #2: Yes

5. Review Comments to the Author

Reviewer #1: I suggest that the title be rewritten. The evaluated method showed limitations when used in treated water; however it showed good results for analyzing fresh water.

Introduction

P.3, L.36 – Include references;

P.4, L.64 – Amphotericin B is an antifungal, correct in the text;

P.4, L.65 – Include dark field microscopy in the sentence;

P.5, L.84 – Include references.

Matherial and methods

P. 12, L.229 - Include geographic coordinates;

Results

P.13, L.243 – 248 - Remove this paragraph, this information has already been given.

Discussion

P.21, L.404 – The authors the authors identified that the pathogenic strain being more tolerant to heat. This is an interesting finding and should be discussed further.

Reviewer #2: The present study is well-written and properly conducted. The Discussion is the section that stands out with an interesting debate about the possible uses of the technique with an impact on public health. I am not sure of the speed of the developed method, so I suggest modify the objective to: investigate the accuracy of a new molecular method for the quantification of potentially viable leptospires in environmental water samples under disinfectants influence.

Minor revisions:

L63-68; L83-84: are not required in the Introduction.

L99 (We implemented ....) - 105 (... animal samples) must be moved to the beginning of the Discussion.

Do not give paragraph on line 96

In Material and Methods, specify the water collection points better. How many points in the Paris city were collected? Did all points receive water treatment included in the city's sanitation?

L371-379: remove from Discussion. This paragraph is not part of the debate.

6. PLOS authors have the option to publish the peer review history of their article (what does this mean?). If published, this will include your full peer review and any attached files.

Reviewer #1: No

Reviewer #2: No

---

## [Author Response · Author response to Decision Letter 0]

4 May 2021

PONE-D-21-09857

Dear Editor,

Thanks for considering our work to be published in your journal. We replied to all the referee remarks and your comments below, in blue.

Sincerely,

Laurent Moulin

Reviewer #1: I suggest that the title be rewritten. The evaluated method showed limitations when used in treated water; however it showed good results for analyzing fresh water.

As suggested, the title was rewritten.

Introduction

P.3, L.36 – Include references;

References added as requested.

P.4, L.64 – Amphotericin B is an antifungal, correct in the text;

Corrected as suggested

P.4, L.65 – Include dark field microscopy in the sentence;

Dark field microscopy has been included in the sentence.

P.5, L.84 – Include references.

References added as requested.

Matherial and methods

P. 12, L.229 - Include geographic coordinates;

Geographical coordinates were added for samples from the Seine river and from the bathing area.

Results

P.13, L.243 – 248 - Remove this paragraph, this information has already been given.

This paragraph was removed as suggested.

Discussion

P.21, L.404 – The authors the authors identified that the pathogenic strain being more tolerant to heat. This is an interesting finding and should be discussed further.

As suggested, this result is now discussed. 

Reviewer #2:

The present study is well-written and properly conducted. The Discussion is the section that stands out with an interesting debate about the possible uses of the technique with an impact on public health. I am not sure of the speed of the developed method, so I suggest modify the objective to: investigate the accuracy of a new molecular method for the quantification of potentially viable leptospires in environmental water samples under disinfectants influence.

Many thanks to the reviewers for these nice comments

Minor revisions:

L63-68; L83-84: are not required in the Introduction.

Lanes 63-68 has been shortened and Lanes 83-84 has been removed, as suggested

L99 (We implemented ....) - 105 (... animal samples) must be moved to the beginning of the Discussion.

As suggested, this part has been moved to the beginning of the discussion.

Do not give paragraph on line 96

Paragraph line 96 has been removed

In Material and Methods, specify the water collection points better. How many points in the Paris city were collected? Did all points receive water treatment included in the city's sanitation?

Geographical localization and number of samples were added to this paragraph. These samples were composed by surface water which had not undergone any sanitation treatment.

L371-379: remove from Discussion. This paragraph is not part of the debate.

Most of the paragraph has been shortened

---

## [Editor Report · Decision Letter 1]

6 May 2021

Effect of disinfection agents and quantification of potentially viable Leptospira in fresh water samples using a highly sensitive integrity-qPCR assay

PONE-D-21-09857R1

Dear Dr. Moulin,

We’re pleased to inform you that your manuscript has been judged scientifically suitable for publication and will be formally accepted for publication once it meets all outstanding technical requirements.

Kind regards,

Odir Antonio Dellagostin

Academic Editor

PLOS ONE
---

## [Editor Report · Acceptance letter]

17 May 2021

PONE-D-21-09857R1 

 Effect of disinfection agents and quantification of potentially viable Leptospira in fresh water samples using a highly sensitive integrity-qPCR assay 

Dear Dr. Moulin:

I'm pleased to inform you that your manuscript has been deemed suitable for publication in PLOS ONE. Congratulations! Your manuscript is now with our production department. 

Kind regards, 

on behalf of

Dr. Odir Antonio Dellagostin 

Academic Editor

PLOS ONE